# Malaria prevention practices and associated factors among households of Hawassa City Administration, Southern Ethiopia, 2020

Anteneh Fikrie [1,2]*, Mihiret Kayamo[2], Henok Bekele[2,3]

1 School of Public Health, College of Health and Medical Sciences, Bule Hora University, Bule Hora, Oromia, Ethiopia, 2 Public Health Department, Pharma College Hawassa Campus, Hawassa, Sidama, Ethiopia, 3 Malaria Prevention, Control and Elimination Program Technical Advisory in South Nation Nationalities Peoples Regional State, Southern Ethiopia

* antenehfikrie3@gmail.com

**Data Availability Statement:** All relevant data are within the paper and supporting information files.

## Abstract

### Background

Despite it is easily preventable; malaria is still remains to be a major public health problem in globally as well as in Ethiopia. The disease can be easily prevented through individual and societal combined efforts by keeping the environment safe, effective utilization of long lasting Insecticide Nets and early treatment. However, the factors for poor knowledge and practices of malaria prevention is not well studied in Ethiopia; particularly, in the study area. Hence, this study aimed to provide concrete evidence towards malaria prevention practices and associated factors among Households of Hawassa City Administration, Southern Ethiopia, 2020.

### Method

A community-based cross-sectional study was conducted among a randomly selected 598 households at Hawassa City Administration from April 1–15, 2020. Multistage sampling technique was employed to recruit the study households. Data were collected by trained data collectors through a face-to-face interview with pretested structured questionnaire, which was adapted from previous peer reviewed articles. Then the data were checked for the completeness and consistencies, then, coded and entered into Epi data 3.1 and it was exported to SPSS IBM version 23 for analysis. Descriptive mean with standard deviation was used to summarize the continuous variables. Bivariable and multivariable logistic regression model was used to assess factors affecting prevention and control of Malaria. Finally, adjusted odds ratio together with 95% CI and p-value <0.05 was used to declare the statistical significances.

### Results

The overall 317 (54.3%) of households practiced good measure of malaria prevention and control measures. Urban residence [AOR = 1.95 (95%CI: 1.17–3.24)], Secondary school completed [AOR = 5.02(95%CI 2.24–12.03)], Tertiary school completed [AOR = 7.27(95%

**Funding:** Pharma College has funded the research. The funder had no role in study design, data collection and analysis, decision to publish, or preparation of the manuscript.

**Competing interests:** I have read the journal's policy and the authors of this manuscript have the following competing interests. The authors declare that they have no competing interests.

CI: 2.84–18.55)], Positive Attitude [AOR = 8.20(95%CI: 5.31–12.68)] and Good knowledge about malaria [AOR = 2.81(95%CI: 1.78–4.44)] were significantly associated with malaria prevention practices.

## Conclusions

Nearly half of the households were still practiced poor measure of malaria prevention and control measures. Hence, health officials and stake holders need attention by providing continuous health education and follow up to control malaria.

## Introduction

Malaria is a life-threatening and an acute infection, of red blood cells caused by protozoa parasites of the genus Plasmodium. The parasites are spread from person to person through the bite of infected female anopheles mosquitoes. There are four parasite species that cause malaria in humans:P. Vivax, P. malariae, P. falciparum and P. ovale. Of this, P.falciparum and P.Vivax poses the greatest risk [1]. It is preventable and curable [2]. All peoples are at risk of contracting malaria, however pregnant women, infants, patients with HIV/AIDS and children under five years old are the most vulnerable to dying of malaria or suffering serious consequences of the disease [1, 3]. Malaria had great impacts the health of the fetus, leading to prematurity and low birth weight [1]. Intensified vector control and routine case management had a differential impact on rates of malaria in young children [4]. In 2018, P.falciparum accounted for 99.7%, 71% of estimated malaria cases in the WHO African Region and WHO South-East Asian Region respectively. Whereas, P.Vivax is the predominant parasite for accounting 75% of malaria cases in the WHO Region of the American [1].

The World malaria report estimated **228 and 229** million malaria cases globally in 2018 and 2019 respectively. Almost 95% of all malaria cases globally were in 29 countries. An estimated number of malaria deaths stood at 405,000 and 409, 000 in 2018 and 2019 respectively. Children aged under 5 years accounted for 67% of all malaria deaths worldwide [5, 6]. In Ethiopia, approximately 60% of the population lives in malaria risk areas and the problem is compounded by increasing frequency and magnitude of malaria epidemics [7]. Moreover, in Ethiopia malaria transmission is unstable, highly seasonal and varies geographically across the country, and outbreaks are considered public health emergencies [8]. More importantly, during the current COVID-19 pandemic, practicing malaria prevention mechanisms have of greatest advantage as the evidence showed the association between COVID-19 and malaria epidemics was devastating particularly in low and middle-income countries like Ethiopia, the regions where malaria is endemic, are at risk of suffering from the consequences of COVID-19 due to mutual side effects, such as less access to treatment for patients with malaria due to the fear of access to healthcare centers leading to worse outcomes and diagnostic delays [9]. The WHO recommended each country to ensure the maintenance of malaria services as part of the essential health package in the country while working to control COVID-19 [10].

In 2017/18, 1,206,892 confirmed and clinical malaria cases and 158 deaths have been reported, of these, nearly 69.2% and 30.8% of cases were Plasmodium falciparum and Plasmodium vivax respectively [11]. According to World Malaria Report, Ethiopia has shown 57 percent decline in incidence and a 54 percent reduction in malaria mortality between 2015 and 2018 [1]. The current, malaria parasite prevalence is 1.3% in areas where the altitude is below 2,000 meters by microscopy for all ages. This was achieved through a rapid scale-up of four

proven and highly effective malaria prevention and treatment measures: insecticide treated mosquito nets (ITNs); indoor residual spraying (IRS); accurate diagnosis and prompt treatment with artemisinin-based combination t herapies (ACTs); and intermittent preventive treatment of pregnant women (IPTp) regions [11].

World Health Organization recommends utilization of ITN and indoor residual spraying (IRS) for preventing mosquito bites. By 2018, 72% of households in sub-Saharan Africa had at least one ITN and about 57% of the population had access to an ITN, while 40% of the population lived in households with enough ITNs for all occupants [1]. Ethiopia has recently declared malaria elimination efforts in 239 selected districts located in six different regions including the study setting, Hawassa City [11]. This effort will be achieved through the implementation of vector control activities such as: indoor residual spraying (IRS), long lasting insecticidal nets (LLINs) and larval source reduction (LSM) [7].

In Ethiopia despite working towards vector control and epidemic prevention as one of the core strategy still significant number of households have poor practice related to malaria prevention [12]. The possible reasons for these poor practices could be due to poor or no awareness and misuse of malaria prevention practices [12, 13]. Moreover, significant numbers of communities are still suffering from malaria morbidity [12, 14–16]. However, factors affecting malaria prevention practice is not well studied in Ethiopia; particularly after the declaration of malaria elimination in the study area. Furthermore, the current Ethiopian national malaria strategic objective stated that: by the year 2020 all households living in malaria endemic area will have the Knowledge, Attitude and Practice used to adopt appropriate health seeking behavior for malaria prevention and control [11]. Therefore, this study provides an answer for the above statement and also identifies the factors affecting malaria prevention practices for policy makers, non-governmental organizations and program managers to improve the strategies.

## Materials and methods

### Study setting, design and period

A community-based cross-sectional study design was carried out in Hawassa City administration from April 1–15, 2020. Hawassa City is serving as a capital city of Southern Nations, Nationalities, and People Region (SNNPR) and newly emerged Sidama National Regional State concomitantly is located 275km south of far from Addis Ababa, Ethiopia. The city has 8 administrative sub cities and segregated into 20 urban and 12 rural kebeles. According to the report of housing and population census, the projected population of Hawassa city Administration in 2018 was estimated to be 367, 908 and comprises about 75,084 households [17]. The city is under the malaria elimination targeted cities nationally. The city has 3 governmental Hospitals, 12 Governmental health Centers and 13 health posts. Further, there are 4 hospital, 2 Specialty clinics, and 44 medium clinics owned by private. The city health department has distributed 74,600 ITNs for the city households. According to the City Health Department report, there were 9971 and 9,217 confirmed malaria cases by the year 2018 and 2019 respectively [18].

### Study population, sample size determination and procedure

The study source populations were all households' lives in Hawassa city. Whereas, the study populations were all randomly selected households lives in the selected sub-cities. Adult's age ≥ 18 years and who lived within the city for at least 6 months duration were included in the study. All adults who were unable to hear or communicate and critically ill individuals at the period of data collection were excluded from the study. The sample size for first objective

was determined using single population proportion formula based on the following assumptions: Proportion (62%) of malaria prevention and control in Gurage Zone [12], 95% Confidence interval 5% margin of error, design effect of 1.5 and 10% for compensation non-response rate. Then, the minimum calculated sample size became 598 Households. In order to identify factors associated with malaria prevention practices, the adequacy of the sample size was examined using Epi Info 7.2 software. Hence, the largest sample size calculated was obtained from first objective as compared to sample size calculated to the second objective. A multistage sampling technique was employed to select the study participants. First, three sub-cities out of eight were selected by using simple random sampling technique lottery method. Second, nine kebeles were selected out of 32 kebeles by using a similar technique. Then the sample was allocated proportionally for the nine randomly selected kebeles. Finally a computer generated simple random sampling technique was employed so as to select individual households. One eligible adult was selected from each household using lottery method for the case where more than one adult was living in the house.

## Data collection procedure and quality assurance

A face-to-face interview administered structured and pretested questionnaire, which was adapted from different peer reviewed articles [12, 19, 20]. The questionnaire was primarily prepared in English version then translated in to local languages, 'Amharic and Sidama Affoo' then it was translated back into English version to check the consistence of the data (S1 File). Four diploma holder Nurses and two BSc degree holder nurses was recruited as data collectors and supervisors respectively. Training was given for data collectors and supervisor for two days on method of data collection through interviewing, how to fill the information on a structured questionnaire. During the data collection period supervisors were checked for the completeness and consistencies of the collected data.

## Study variables and operational definition

The dependent variable of the study was practice of malaria prevention and the explanatory variables were Socio demographic Factors (Age, Sex, Marital status, Educational level, Occupation, income), Knowledge and attitude about malaria.

**Knowledge.** Participants who answered ≥50% of correct answers among the total knowledge related questions were classified as having a good knowledge; whereas participants who answered < 50% of the questions were classified as having poor knowledge [12].

**Malaria prevention practice.** Participants who answered ≥50% of correct answers among the total malaria prevention practice related questions were classified as having a good practice. While participants who answered < 50% of the questions were classified as having poor practice [12].

**Attitude.** Participants who answered ≥50% of correct answers among the total attitude related questions were classified as having a positive attitude. Whereas, participants who answered <50% of questions were classified as having negative attitude [12].

## Data processing and analysis

At the very beginning, completeness and consistencies of the data were checked then data template format was prepared and entered into Epidat 3.1 and it was exported to SPSS version 23 for analysis. Descriptive statics was run using central tendency (mean) and dispersion (Standard deviation) for continuous variables and also computed for frequency and percentages for categorical variables. The data were presented by tables. The bivariate and multivariate logistic regression was used to identify the possible factors affecting the practice of malaria prevention

among Households of Hawassa city. A variable with a p-value of $\leq 0.25$ during bivariate analysis was entered in to multivariate logistic regression for further analysis to control the effect of confounding variables. Hosmer and Lemshow test static were used to check the assumption of logistic regression. Adjusted odds ratio with 95% confidence interval and p value <0.05 was used to declare the significance of all statistic.

### Ethics approval and consent to participate

Primarily an ethical clearance letter was obtained from Institutional Review Board of Pharma College, School of graduate studies department of Public Health. Additional permission letter was also be secured from Hawassa city administrations health Department and from all selected kebeles administrative offices. All participants were offered with adequate information regarding the purpose, risk and benefit, and confidentiality of the study prior to the data collection. Participation was fully voluntary and verbal informed consent was obtained from each participant. The ethics committees had approved the verbal consent procedure. Code numbers was used in place of identifiers to maintain the confidentiality of participants' information.

## Results

### Socio-demographic characteristics of study participants

Out of the total 598 selected participants, 584 of them were voluntarily interviewed and yielding a response rate of 97.7%. The mean (± SD) age of participant was 34.7 (± 9.69) and the majority, 330 (56.5%) and 413 (70.7%) of participants were male and employed respectively (Table 1).

### Knowledge about malaria prevention among study participants

Of the total, 453 (77.6%) of participants knew the cause of malaria. The vast majority, 547 (93.7%) of the participants knew water body was the common breeding sites for the mosquito. Regarding the knowledge level; nearly two-third (63.9%) [95% CI: 59.9% - 67.9] of the participants have good knowledge towards malaria causes, transmission, sign and symptoms and prevention measures (Table 2).

### Attitude towards malaria prevention and control among households

More than the half of the respondents, 325 (55.7%) perceived that Malaria is a life-threatening disease. Likewise, more than half of the respondents 311 (53.3%) believed that sleeping under a mosquito net is one of the methods of malaria prevention. Whereas, the majority, 310 (53.1%) of respondents perceive that self-treatment can endangers one's health. The total score of attitude showed that, more than half 57.5% [95% CI: 53.6–61.6] of the participants have positive attitudes towards the natures of malaria (Table 3).

### Practice of malaria prevention and control among households

A total of 584 of respondents, who used of malaria prevention methods always, 245 (42.0%) use mosquito repellents, 246 (42.1%) use anti-mosquito spray and 237 (40.6%) sprayed with anti-mosquito chemical spray (IRS). Above the one third of the respondents, 257 (44.0%) were always sleep in a mosquito net, and 231 (39.6%) were of other members of the household sleep in mosquito nets. Regarding to the handling of the ITNs, 248 (42.5%) and 240 (41.1%) were always practice clean/cut bushes and clean stagnant water near their house, respectively. Among all households, 177 (30.3%) were always visits the health center when they feel sick, while 145 (24.8%) were received from the health extension worker and 121 (20.7%) were

**Table 1. Socio-demographic and economic characteristics of respondents at Hawassa City, South Ethiopia, 2020.**

| Variable | Category | Frequency | Percent % |
|---|---|---|---|
| Age in years | ≤24 | 99 | 17 |
| | 25–34 | 237 | 40.6 |
| | 35–44 | 139 | 23.8 |
| | ≥45 | 109 | 18.7 |
| Sex | Female | 330 | 56.5 |
| | Male | 254 | 43.5 |
| Residence | Rural | 169 | 28.9 |
| | Urban | 415 | 71.1 |
| Marital status | Married | 553 | 94.7 |
| | Single | 9 | 1.5 |
| | Divorced | 19 | 3.3 |
| | Widowed | 3 | .5 |
| Educational status | No formal education | 68 | 11.6 |
| | Primary completed | 183 | 31.3 |
| | Secondary completed | 197 | 33.7 |
| | Tertiary completed | 136 | 23.3 |
| Household monthly income | ≤ 1000 ETB | 180 | 30.8 |
| | 1001–2575 ETB | 181 | 31.0 |
| | > 2575 ETB | 223 | 38.2 |
| Occupational status | House wife | 66 | 11.3 |
| | Farmer | 94 | 16.1 |
| | Employed | 413 | 70.7 |
| | Merchants | 11 | 1.9 |

participated in malaria prevention campaigns. Regarding the overall malaria prevention practice; more than half 317 [(54.3%), 95% CI: 50.2% - 58.3%] of households practiced good measure of malaria prevention (Table 4).

## Factors associated with good practice of malaria prevention and control

After controlling the potentials confounding variables by the multivariate analysis: Urban residence [AOR = 1.95 (95%CI: 1.17–3.24)], Secondary school completed [AOR = 5.02(95%CI 2.24–12.03)], Tertiary school completed [AOR = 7.27(95%CI: 2.84–18.55)], Positive Attitude [AOR = 8.20(95%CI: 5.31–12.68)], Good knowledge about malaria [AOR = 2.81(95%CI: 1.78–4.44)] were significantly associated with malaria prevention practices. Accordingly, the odds of good malaria prevention practice were nearly 2 times higher among urban residence of study participants as compared to the rural resident [AOR = 1.95 (95%CI: 1.17–3.24)]. Those participants with an educational level of secondary school completed had 5 times more likely to have good malaria prevention practices than those who do not have formal education [AOR = 5.02 (95%CI 2.24–12.03)]. Similarly, those participants with educational level of tertiary school completed were 7 times more likely of having a good malaria prevention practices as compared to those who do not have formal education [AOR = 7.27(95%CI: 2.84–18.55)]. A study participants who have positive attitude about malaria were nearly 8 times more likely of having good malaria prevention practices as compared to those study participants who have negative attitude [AOR = 8.20(95%CI: 5.31–12.68)]. Likewise, a study participants who have good knowledge about malaria were nearly 3 times more likely of having good malaria prevention

**Table 2. Knowledge about malaria prevention and control among households in Hawassa City, South Ethiopia, 2020 (n = 584).**

| Variable | Category | No. | Percent % |
|---|---|---|---|
| Cause of malaria was parasites | Yes | 453 | 77.6 |
| | No | 131 | 22.4 |
| Signs/symptoms of malaria | Fevered | 310 | 53.1 |
| | Chills | 134 | 22.9 |
| | Headache | 67 | 11.5 |
| | Joint pain | 64 | 11.0 |
| | Vomiting | 9 | 1.5 |
| Mosquito bite mostly during | Day | 109 | 18.7 |
| | Night | 475 | 81.3 |
| Common breeding sites | Dry area | 37 | 6.3 |
| | Water body | 547 | 93.7 |
| Common resting sites | House | 506 | 86.6 |
| | Outside house | 78 | 13.4 |
| Mode of transmission | Mosquito bite | 554 | 94.9 |
| | Fly bite & Drinking water | 30 | 5.1 |
| Prevention methods of malaria | ITN use | 571 | 97.8 |
| | Drainage of stagnant water | 529 | 90.6 |
| | Covering body parts | 259 | 44.3 |
| | Repellant use | 495 | 84.8 |
| | Close openings | 323 | 55.3 |
| Advantage of ITN | Prevent mosquito bite | 558 | 95.6 |
| | Attract mosquito | 26 | 4.4 |
| Most at risk group affected by malaria | Pregnant & children | 356 | 61 |
| | Others | 228 | 39 |
| Knowledge | Good | 373 | 63.9 |
| | Poor | 211 | 36.1 |

practices as compared to those study participants who have poor knowledge about malaria [AOR = 2.81(95%CI: 1.78–4.44)] (Table 5).

## Discussions

According to the WHO recommendations each nation's ministry of health must ensure that malaria control efforts are not hampered or neglected as they tackle the COVID-19 pandemic [10]. Malaria and COVID-19 may have similar aspects and seem to have a strong potential for mutual influence. They have already caused millions of deaths, and the regions where malaria is endemic are at risk of suffering from the consequences of COVID-19 due to mutual side effects, such as less access to treatment for patients with malaria due to the fear of access to healthcare centers leading to worse outcomes and diagnostic delays. Moreover, the similar and generic symptoms make it harder to achieve an immediate diagnosis [9]. Hence, this community-based cross-sectional study was assessed Malaria prevention practices and associated factors among Households of Hawassa City Administration, Southern Ethiopia during the surge of COVID-19. Accordingly, the study result revealed that the overall malaria prevention practice of the study participants is found to be 54.3% [95% CI, (50.2–58.3)]. Being urban resident, secondary and tertiary school completion, having positive attitude and good knowledge about malaria were positively associated with malaria prevention practices.

**Table 3. Attitude towards malaria prevention and control among households in Hawassa City, South Ethiopia, 2020.**

| Variables | Disagree | | Not sure | | Agree | |
|---|---|---|---|---|---|---|
| | No. | (%) | No. | (%) | No. | (%) |
| Malaria is a fatal disease | 144 | (24.7) | 115 | (19.7) | 325 | (55.7) |
| Malaria is a communicable disease | 143 | (24.5) | 114 | (19.5) | 327 | (56.0) |
| The best way to protect myself getting from Malaria is avoiding mosquito bites | 141 | (24.1) | 112 | (19.2) | 331 | (56.7) |
| Anyone can get Malaria | 152 | (26.0) | 125 | (21.4) | 307 | (52.6) |
| Sleeping under a mosquito net during the night is prevent from Malaria | 141 | (24.1) | 132 | (22.6) | 311 | (53.3) |
| Self-treatment may dangers one's health | 134 | (22.9) | 140 | (24.0) | 310 | (53.1) |
| Children and pregnant women are at higher risk of getting Malaria | 147 | (25.2) | 126 | (21.6) | 311 | (53.3) |
| Malaria recover without any treatment | 132 | (22.6) | 142 | (24.3) | 310 | (53.1) |
| Malaria can't transmit through contact with others | 142 | (24.3) | 109 | (18.7) | 333 | (57.0) |
| Working/ sitting at night time in the outside had greater risk of getting Malaria | 144 | (24.7) | 126 | (21.6) | 314 | (53.8) |
| When Malaria medicine is not taken completely, bad consequences will be resulted | 145 | (24.8) | 126 | (21.6) | 313 | (53.6) |
| Anti-Malaria drugs can be easily purchased from the drug store/pharmacy so as to treat myself when I get Malaria | 140 | (24.0) | 119 | (20.4) | 325 | (55.7) |
| I think that I should have a blood test if i will have a fever | 139 | (23.8) | 130 | (22.3) | 315 | (53.9) |
| I will seek for advice in case I have contracted Malaria | 146 | (25.0) | 116 | (19.9) | 322 | (55.1) |
| Checking the expiry date of anti-malaria drug is very important before taking it. | 158 | (27.1) | 109 | (18.7) | 317 | (54.3) |
| Attitude | | | | | | |
| Negative | | | 248 (42.5%) | | | |
| Positive | | | 336 (57.5%) | | | |

The prevalence of practice of malaria prevention measures obtained in this study is consistent with previous study reports 53.5% in in Central Tanzania [21] and 55% in Southwestern Ethiopia [14]. This may due to the study participants in the Africa had almost had at similar socio-economic status and the implementation of the practice of malaria prevention and control measures at the community level. However, the result of this study is higher as compared with the previous studies report from Nigeria [22] and Cameroon [23]. On contrary, the study prevalence of practice of malaria prevention measures found in this study is lower than previous studies conducted in Ethiopia [7, 16, 17]. This discrepancy might be due to varied intense intervention implemented by the government health authorities of the region.

This study found that nearly three-fourth, 73.5% of study participants are sleeping under an ITN. The highest percentage, 68% of ITN utilization was obtained in pregnant and children. This is slightly lower than the targets of national operational plan of malaria, which was

**Table 4. Practice of malaria prevention among households at in Hawassa City, South Ethiopia, 2020.**

| Variables | Never | | Sometimes | | Always | |
|---|---|---|---|---|---|---|
| | No. | (%) | No. | (%) | No. | (%) |
| How frequently do you sleep in ITN | 155 | (26.5) | 172 | (29.5) | 257 | (44.0) |
| How frequently others members you family sleep in ITN | 139 | (23.8) | 214 | (36.6) | 231 | (39.6) |
| How frequently you used mosquito repellents | 133 | (22.8) | 206 | (35.3) | 245 | (42.0) |
| How frequently you used anti-mosquito spray in your house | 141 | (24.1) | 197 | (33.7) | 246 | (42.1) |
| How frequently your house sprayed with anti-mosquito chemical spray (IRS) | 136 | (23.3) | 211 | (36.1) | 237 | (40.6) |
| How frequently do you clean/cut bushes surrounding your house | 148 | (25.3) | 188 | (32.2) | 248 | (42.5) |
| How frequently do you clean the stagnant water near in your house | 134 | (22.9) | 210 | (36.0) | 240 | (41.1) |
| How frequently do you visit Health Center Hospital/ Clinic when you feel sick | 212 | (36.3) | 195 | (33.4) | 177 | (30.3) |
| How frequently did you receive home-to-home visits by the nearby health extension workers | 298 | (51.0) | 141 | (24.1) | 145 | (24.8) |
| How frequently did you participated in malaria prevention campaigns | 319 | (54.6) | 144 | (24.7) | 121 | (20.7) |

**Table 5. Bivariable and multivariable logistic regression analysis for malaria prevention practice and associated factors among households in Hawassa City, South Ethiopia, 2020.**

| Variable | | Malaria prevention practice | | | | COR (95% Cl) | AOR (95% Cl) |
|---|---|---|---|---|---|---|---|
| | | Good | | Poor | | | |
| | | No. | (%) | No. | (%) | | |
| Sex | | | | | | | |
| | Male | 127 | 50.0 | 127 | 50.0 | 1 | 1 |
| | Female | 190 | 57.6 | 140 | 42.4 | 1.4(0.97, 1.9) | 1.33(0.87, 2.03) |
| Age | | | | | | | |
| | ≤24 | 39 | 39.4 | 60 | 60.6 | 0.66(0.38–1.14) | 0.53(0.25–1.12) |
| | 25–34 | 146 | 61.6 | 91 | 38.4 | 1.63(1.03–2.58) | 1.22(0.65–2.29) |
| | 35–44 | 78 | 56.1 | 61 | 43.9 | 1.30(0.78–2.15) | 1.02(0.53–1.96) |
| | ≥45 | 54 | 49.5 | 55 | 50.5 | 1 | 1 |
| Address | | | | | | | |
| | Urban | 246 | 59.3 | 169 | (40.7) | 2.01(1.4, 2.88) | 1.95(1.17, 3.24)** |
| | Rural | 71 | 42.0 | 98 | (58.0) | 1 | 1 |
| Educational status | | | | | | | |
| | No formal education | 18 | 26.5 | 50 | (73.5) | 1 | 1 |
| | Primary Completed | 77 | 42.1 | 106 | (57.9) | 2.02(1.09, 3.7) | 1.81(0.83, 3.95) |
| | Secondary Completed | 131 | 66.5 | 66 | (33.5) | 5.51(3.09, 9.9) | 5.02(2.24, 12.03)*** |
| | Tertiary Completed | 91 | 69.9 | 45 | 29.1 | 5.61(2.94,10.72) | 7.27(2.84,18.55)*** |
| Household monthly income | | | | | | | |
| | ≤ 1000 ETB | 78 | (43.3) | 102 | (56.7) | 1 | 1 |
| | 1001–2575 ETB | 98 | (54.1) | 83 | (45.9) | 1.5(1.02, 2.3) | 1.02(0.59, 1.76) |
| | > 2575 ETB | 141 | (63.2) | 82 | (36.8) | 2.2(1.5, 3.4) | 1.57(0.94, 2.61) |
| Attitude towards malaria | | | | | | | |
| | Negative | 58 | (23.4) | 190 | (76.6) | 1 | 1 |
| | Positive | 259 | (77.1) | 77 | (22.9) | 11.02(7.5, 16.3) | 8.20(5.31, 12.68)** |
| Knowledge about malaria | | | | | | | |
| | Poor | 82 | (38.9) | 129 | (61.1) | 1 | 1 |
| | Good | 235 | (63.0) | 138 | (37.0) | 2.6(1.8, 3.7) | 2.81(1.78, 4.44)*** |
| Occupational status | | | | | | | |
| | Employed | 249 | 60.3 | 164 | 39.7 | 2.3(1.58–3.31) | 0.73(0.41–1.30) |
| | Non-employed | 68 | 39.8 | 103 | 60.2 | 1 | 1 |

NB; ETB-Ethiopian Birr(1ETB = 0.0308$)

* P-value < 0.05

** P-value < 0.01

*** P-value < 0.001 was statistically significant, 1 = Reference.

planned to have the levels of above 80%. This could be due to the movement of peoples in to the city from the nearby towns and districts and this result scarcity of ITN at the household level. On the other hand this is comparable with a study conducted at Southern Ethiopia [8]. However, this finding is by far higher than a report of sub-Saharan Africa population at risk of malaria [1, 19]. The possible explanation for the observed higher utilization rate could due to the study areas which were included for malaria sub-national elimination program.

On the other hand, more than three-fourth, 76.7% of study participants are protected by IRS. This is comparably higher than study conducted at Southern Ethiopia [8, 19, 24]. However, higher than a study done at Cape verdea where only 42.9% of households were sprayed [25]. The reason behind this scenario could be that in most countries, IRS is targeted at a few

focal or targeted areas, which may vary over time. Likewise, the study area is surrounded with Lake Hawassa, so that this would also be the reason for the observed high coverage of IRS. More than three-fourth, of the study participants were actively participated in drainage of stagnant water and cleaning bushes surrounding their house. This might be achieved through strong leadership, participatory and community inclusive implementation of interventions.

The study participants from urban resident were nearly 2 times more likely to have good practice of malaria prevention than rural residents. This is in line with a study report from Cameroon [23], Southern Tanzania [26] and Malawi [27]. This may due to the reason that urban dwellers are more exposed to different source of information which helps to lift the participant's knowledge about malaria preventive methods and benefits of sleeping under ITNs. This study found that good malaria prevention practice by households was increased with increasing education level of participants. Thus, the odd of good malaria prevention practice was higher among secondary and tertiary school completed participants than those who do not have formal education. In line to this finding priors research results concluded the same [14, 15, 21]. This might be due to the reason that higher educational status accompanied with high income; which increases once purchasing power of mosquito nets individually rather than awaiting from the government.

Study participants who have good knowledge and positive attitude about malaria were more likely of having good malaria prevention practices as compared to their counter parts. This result was supported by the study report from Southwestern Ethiopia [14] and Malawi [27]. This could be justified as the level of knowledge and attitude regarding the mode of transmission, risk factor, seriousness of the disease and mechanism of prevention of the individual increases, the probability of practicing preventive activities will also be elevated. Moreover, having good knowledge and positive attitude towards the malaria prevention measures triggers once motives of practice.

This community based study has benefited from high response rate with relatively large calculated sample size which will enhance the generalizability of the study results. However, the study has its own limitation, particularly the nature of the study design employed, and the presence of social desirability bias which may overestimate or underestimate the results of malaria prevention practice and control measures.

## Conclusion

The overall malaria prevention practice of the study participants is found to be 54.3%. Though the practice of malaria prevention measures obtained in this study was comparable to prior studies, but it is lower than the predetermined strategic objective of Ethiopian operational plan for malaria. The study participants who are sleeping under an ITN showed slightly lower than the targets of national operational plan of malaria, which was planned to have the levels above 80%. More than three-fourth, 76.7% of study participants are protected by IRS. More than three-fourth, of the study participants were actively participated in drainage of stagnant water and cleaning bushes surrounding their house. Being urban resident, secondary and tertiary school completion, having positive attitude and good knowledge about malaria were positively associated with malaria prevention practices.

Therefore, the local as well as the regional government should give more emphasis on the regular supervision and support with the capacity of the health education and promotion services to maximize the good the practice ITN utilization. It needs also to maximize the knowledge and attitude of communities towards malaria prevention and control measures through behavior change communication media that are accessible and appropriate to illiterate groups of populations. Further mixed study should be conducted at community level to identify the

determinants of poor knowledge and negative attitude of participants towards malaria prevention practice measure.

## Supporting information

**S1 File. English version questionnaire.**
(DOCX)

**S2 File. Raw SPSS dataset.**
(SAV)

## Acknowledgments

We would like to express our sincerely gratitude to the Hawassa City Administration for permitting us to undertake this study. Last but not least, a big appreciation goes to our data collectors, supervisors and those who were actively participated in our study.

## Author Contributions

**Conceptualization:** Anteneh Fikrie, Mihiret Kayamo.

**Data curation:** Anteneh Fikrie, Mihiret Kayamo.

**Formal analysis:** Anteneh Fikrie, Mihiret Kayamo.

**Funding acquisition:** Anteneh Fikrie, Mihiret Kayamo.

**Investigation:** Anteneh Fikrie, Mihiret Kayamo.

**Methodology:** Anteneh Fikrie, Mihiret Kayamo.

**Project administration:** Anteneh Fikrie, Mihiret Kayamo.

**Resources:** Anteneh Fikrie, Mihiret Kayamo.

**Software:** Anteneh Fikrie, Mihiret Kayamo.

**Supervision:** Anteneh Fikrie, Mihiret Kayamo, Henok Bekele.

**Validation:** Anteneh Fikrie, Mihiret Kayamo, Henok Bekele.

**Visualization:** Anteneh Fikrie, Mihiret Kayamo, Henok Bekele.

**Writing – original draft:** Anteneh Fikrie, Mihiret Kayamo, Henok Bekele.

**Writing – review & editing:** Anteneh Fikrie, Mihiret Kayamo, Henok Bekele.

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
