## [Decision Letter · Decision Letter 0]

26 Mar 2021

PONE-D-20-39939

Malaria Prevention practices and associated factors among Households of Hawassa City Administration, Southern Ethiopia, 2020

PLOS ONE

Dear Dr. Fikrie,

Thank you for submitting your manuscript to PLOS ONE. After careful consideration, we feel that it has merit but does not fully meet PLOS ONE’s publication criteria as it currently stands. Therefore, we invite you to submit a revised version of the manuscript that addresses the points raised during the review process.

We look forward to receiving your revised manuscript.

Kind regards,

Francesco Di Gennaro

Academic Editor

PLOS ONE

Additional Editor Comments:

dear Authors follow reviewer suggestions to improve your paper

Journal Requirements:

2. Please provide additional details regarding participant consent. In the ethics statement in the Methods and online submission information, please describe how verbal consent was documented and witnessed, and why written consent was not obtained. If your study included minors, state whether you obtained consent from parents or guardians.

3. Please include a copy of the interview guide used in the study, in both the original language and English, as Supporting Information, or include a citation if it has been published previously.

4a) If there are ethical or legal restrictions on sharing a de-identified data set, please explain them in detail (e.g., data contain potentially identifying or sensitive patient information) and who has imposed them (e.g., an ethics committee). Please also provide contact information for a data access committee, ethics committee, or other institutional body to which data requests may be sent.

4b) If there are no restrictions, please upload the minimal anonymized data set necessary to replicate your study findings as either Supporting Information files or to a stable, public repository and provide us with the relevant URLs, DOIs, or accession numbers. Please see http://www.bmj.com/content/340/bmj.c181.long for guidelines on how to de-identify and prepare clinical data for publication. For a list of acceptable repositories, please see http://journals.plos.org/plosone/s/data-availability#loc-recommended-repositories.

Reviewers' comments:

Reviewer's Responses to Questions

**Comments to the Author**

1. Is the manuscript technically sound, and do the data support the conclusions?

Reviewer #1: Yes

Reviewer #2: Partly

2. Has the statistical analysis been performed appropriately and rigorously? 

Reviewer #1: Yes

Reviewer #2: Yes

3. Have the authors made all data underlying the findings in their manuscript fully available?

Reviewer #1: Yes

Reviewer #2: Yes

4. Is the manuscript presented in an intelligible fashion and written in standard English?

Reviewer #1: Yes

Reviewer #2: Yes

5. Review Comments to the Author

Reviewer #1: Authors worte an interesting paper on Malaria prevention from high setting of Malaria (Ethiopia). I find the paper well wrote and with good statistical analisys

Some minor suggestions:

1. Introduction: Updata data on Malaria burden with Global Malaria report 2020. Furthermore, add some sentence on why also during COVID 19 pandemic is important Malaria control (see and cite Malaria and COVID-19: Common and Different Findings. Trop Med Infect Dis. 2020 Sep 6;5(3):141. doi: 10.3390/tropicalmed5030141. )

2. Methods and result: are clear

3. Discussion: add some sentence on COVID 19 to contexualaize your paper. In fact,Malaria and COVID-19 may have similar aspects and seem to have a strong potential for mutual influence. They have already caused millions of deaths, and the regions where malaria is endemic regions are at risk of suffering from the consequences of COVID-19 due to mutual side effects, such as less access to treatment for patients with malaria due to the fear of access to healthcare centers leading to worse outcomes and diagnostic delays. Moreover, the similar and generic symptoms make it harder to achieve an immediate diagnosis. Healthcare systems and professionals will face a great challenge in case of a syndemic ... etc (see and cite Malaria and COVID-19: Common and Different Findings. Trop Med Infect Dis. 2020 Sep 6;5(3):141. doi: 10.3390/tropicalmed5030141 ) and discuss better on predictors factor of severit in malaria patients (see and if you want cite Prevalence and Predictors of Malaria in Human Immunodeficiency Virus Infected Patients in Beira, Mozambique. Int J Environ Res Public Health. 2018 Sep 17;15(9):2032. doi: 10.3390/ijerph15092032. )

Conclusion : are coherent

Table are well done and table 5 on statistical analisys is clear

Reviewer #2: Abstract

Background: line 1 should read ‘’Despite it is easily preventable, malaria still remains a major…..’’

Introduction:

Page 2 Line 1 should read ‘’have been reported, of these, nearly 69.2%...’’

Page 2 Line 4/5 ‘’in areas below 2,000 meters’’ 2,000 meters below what? this is not clear, clarify

Line 10 ‘’WHO is highly recommends’’ corrected needed here

Method

Data Processing and analysis: 3rd sentence says ‘’the data were presented by tables, graphs and different interactive charts’’ but you only have tables in your manuscript

Results Your result reporting includes both a narrative (running commentary) of the findings and tables. Giving a complete narration of the results and having a table is a bit burdensome. Either go with narrative of the result without a table or give highlight/summary of the result for each table then refer readers to the appropriate table. This is so for tables 2 to 5. Some of the tables are not necessary as a narration of the findings will do just fine.

Knowledge about malaria prevention among study participants: page 8

line 5 need correction of spelling of fever.

Line 6 ‘…participants were knew’’ should be ‘participants knew’.

Line 7 ‘’571 (97.8%) were ever used’’ should be ‘ever used’

Line 8: ‘‘Regarding to the’’ should be ‘regarding the’

Attitude towards Malaria prevention and control among households:page 9

‘’More than the half of the respondents were agreed on 325 (55.7%) were perceived that Malaria is a life-threatening disease, 327 (56.0%) were think that Malaria is a communicable disease and 331 (56.7%) were perceived that the best way to prevent myself getting Malaria is to avoid getting mosquito bites.’’ Please recast as too many use of the word WERE renders this statement meaningless. This also applies to the rest of your results where you used the word WERE a lot, it makes what you are writing difficult to understand

Discussion: it should be discussion, but you have written ‘discussions’

Reference: I suggest authors look up a similar study and add to their references

DePina, A.J., Dia, A.K., de Ascenção Soares Martins, A. et al. Knowledge, attitudes and practices about malaria in Cabo Verde: a country in the pre-elimination context. BMC Public Health 19, 850 (2019). https://doi.org/10.1186/s12889-019-7130-5

6. PLOS authors have the option to publish the peer review history of their article (what does this mean?). If published, this will include your full peer review and any attached files.

Reviewer #1: No

Reviewer #2: **Yes: **Francis Akor

---

## [Author Response · Author response to Decision Letter 0]

7 Apr 2021

29 March, 2021

Dear respected reviewers, 

Ref: A point-by-point response to the comments

Dear reviewer 1, First of all I would like to thank you very much for your appreciation and forwarding constructive and insightful comments which we believe that it would improve the quality of our manuscript. Thus, according to your comments, suggestions and questions the following amendments has been taken.

Amendments done on the minor comments raised by reviewer-1

INTRODUCTION: Update data on Malaria burden with Global Malaria report 2020. 

Answer: I have updated the burden of malaria based on Global Malaria report 2020

Furthermore, add some sentence on why also during COVID 19 pandemic is important Malaria control (see and cite Malaria and COVID-19: Common and Different Findings. Trop Med Infect Dis. 2020 Sep 6;5(3):141. doi: 10.3390/tropicalmed5030141. )

Answer: I have downloaded the aforementioned article and included in our reference list after extracting valuable data’s. 

Discussion: add some sentence on COVID-19 to contextualize your paper

Answer: I have added a sentence describing the existence of COVID-19 and malaria based on your suggestion. 

Amendments done on the comments raised by reviewer-2

Dear Reviewer 2: It is my pleasure to extend my earnest gratitude for your mindful and deep comments and suggestions given to our manuscript. We highly believe that your comments have a big share in the improvement of our manuscript. Here are the amendments taken based on your comments.

Abstract

Background: line 1 should read ‘’Despite it is easily preventable, malaria still remains a major…..’’

Answer: Corrected

Introduction:

Page 2 Line 1 should read ‘’have been reported, of these, nearly 69.2%...’’

Answer: Corrected

Page 2 Line 4/5 ‘’in areas below 2,000 meters’’ 2,000 meters below what? this is not clear, clarify

Answer: ’in areas below 2,000 meters altitude’’

Line 10 ‘’WHO is highly recommends’’ corrected needed here

Answer: Corrected

Methods section

Data Processing and analysis: 3rd sentence says ‘’the data were presented by tables, graphs and different interactive charts’’ but you only have tables in your manuscript

Answer: ‘’the data were presented by tables, graphs and different interactive charts’’ changed to …”the data were presented by tables”. 

Results

Your result reporting includes both a narrative (running commentary) of the findings and tables……. give highlight/summary of the result for each table then refer readers to the appropriate table

Answer: Based on your suggestions I have given highlight of the result for each table.

Knowledge about malaria prevention among study participants: page 8

Line 6 ‘…participants were knew’’ should be ‘participants knew’.

Answer: Corrected!!

Line 7 ‘’571 (97.8%) were ever used’’ should be ‘ever used’

Answer: Corrected!!

Line 8: ‘‘Regarding to the’’ should be ‘regarding the’

Answer: Corrected!!

Attitude towards Malaria prevention and control among households: page 9

Please recast as too many use of the word WERE renders this statement meaningless. This also applies to the rest of your results where you used the word WERE a lot, it makes what you are writing difficult to understand

Answer: Corrected!!

Discussion: it should be discussion, but you have written ‘discussions’

Answer: discussions changed in to ‘discussion’

Reference: I suggest authors look up a similar study and add to their references

Answer: I have added the suggested article in our reference list.

---

## [Decision Letter · Decision Letter 1]

19 Apr 2021

Malaria Prevention practices and associated factors among Households of Hawassa City Administration, Southern Ethiopia, 2020

PONE-D-20-39939R1

Dear Dr Fekrie,

We’re pleased to inform you that your manuscript has been judged scientifically suitable for publication and will be formally accepted for publication once it meets all outstanding technical requirements.

Kind regards,

Francesco Di Gennaro

Academic Editor

PLOS ONE

Additional Editor Comments (optional):

congratulations

Reviewers' comments:

Reviewer's Responses to Questions

**Comments to the Author**

1. If the authors have adequately addressed your comments raised in a previous round of review and you feel that this manuscript is now acceptable for publication, you may indicate that here to bypass the “Comments to the Author” section, enter your conflict of interest statement in the “Confidential to Editor” section, and submit your "Accept" recommendation.

Reviewer #1: All comments have been addressed

Reviewer #2: All comments have been addressed

2. Is the manuscript technically sound, and do the data support the conclusions?

Reviewer #1: Yes

Reviewer #2: (No Response)

3. Has the statistical analysis been performed appropriately and rigorously? 

Reviewer #1: Yes

Reviewer #2: (No Response)

4. Have the authors made all data underlying the findings in their manuscript fully available?

Reviewer #1: Yes

Reviewer #2: (No Response)

5. Is the manuscript presented in an intelligible fashion and written in standard English?

Reviewer #1: Yes

Reviewer #2: (No Response)

6. Review Comments to the Author

Reviewer #1: Authors improved thier paper. The research question is correct also the methods and the setting are relevant

I suggest to accept it

Reviewer #2: (No Response)

7. PLOS authors have the option to publish the peer review history of their article (what does this mean?). If published, this will include your full peer review and any attached files.

Reviewer #1: No

Reviewer #2: **Yes: **Francis Akor

---

## [Editor Report · Acceptance letter]

4 May 2021

PONE-D-20-39939R1 

Malaria Prevention practices and associated factors among Households of Hawassa City Administration, Southern Ethiopia, 2020 

Dear Dr. Fikrie:

I'm pleased to inform you that your manuscript has been deemed suitable for publication in PLOS ONE. Congratulations! Your manuscript is now with our production department. 

Kind regards, 

on behalf of

Dr. Francesco Di Gennaro 

Academic Editor

PLOS ONE